# Tribological Behavior of As-Cast and Aged AlCoCrFeNi$_{2.1}$ CCA

**Fevzi Kafexhiu \*, Bojan Podgornik and Darja Feizpour** 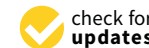

Institute of Metals and Technology, Lepi pot 11, 1000 Ljubljana, Slovenia; bojan.podgornik@imt.si (B.P.); darja.feizpour@imt.si (D.F.)

**\*** Correspondence: fevzi.kafexhiu@imt.si; Tel.: +386-1-4701-931

**Abstract:** In the present study, wear behavior as a function of aging time was evaluated for the AlCoCrFeNi$_{2.1}$ eutectic complex, concentrated alloy (CCA) consisting of B2 (BCC), and L1$_2$ (FCC) lamellae in the as-cast state. By aging the material at 800 °C up to 500 h, precipitation of a fine, evenly dispersed micro-phase inside the L1$_2$ takes place. From 500 h to 1000 h of aging, precipitates coarsen by the Ostwald ripening mechanism. Reciprocating wear tests were characterized by a prevailing abrasive wear mechanism, while adhesive and delamination wear components change with aging conditions. The L1$_2$ phase with lower hardness in the as-cast material preferentially deformed during the wear test, which was not the case after aging the material, i.e., with the presence of precipitates. Aging-induced changes show a similar trend for the coefficient of friction and L1$_2$ + precipitates phase fraction, whereas changes in specific wear rate are in a good agreement with changes in B2 phase fraction. In general, aging the AlCoCrFeNi$_{2.1}$ CCA at 800 °C up to 500 h decreases its coefficient of friction due to reduced adhesive wear component and enhances its wear performance through precipitation strengthening.

**Keywords:** AlCoCrFeNi$_{2.1}$; CCA; HEA; aging; precipitates; wear; tribology

## 1. Introduction

Multi-principal element alloys (MPEAs) have become an attractive topic in the materials science community because of their great potential in discovering and developing new materials of scientific significance and practical benefit. In contrast to conventional alloys, these materials usually contain at least five elements in equimolar or near-equimolar proportions, which in the case of fulfilling the Hume-Rothery rules can have microstructures of single-phase solid solution with high configurational entropy of mixing—high-entropy alloys (HEAs), or they can have complex microstructures that contain multiple phases of solid solution and/or intermetallic compounds—complex, concentrated alloys (CCAs). The research interest on CCAs is ever increasing as they offer an excellent combination of advantages of single-phase solid solution in HEAs, and secondary-phase strengthening effects of well-established alloys. This combination may as well diminish the strength-ductility competition, which is a well-known issue in conventional alloys.

However, the application of these materials in structural engineering requires, among others, a good understanding of their surface degradation mechanisms including corrosion, erosion, and wear behavior [1]. In this respect, some research has been carried out by different authors on different HEAs and CCAs as reviewed by Ayyagari et al. [2]. Some highlights worth mentioning include the work of Wu et al. [3] who found out that by increasing aluminum content of the Al$_x$CoCrCuFeNi, both the volume fraction of the BCC phase and the hardness value increase and thus the wear coefficient decreases. On the other hand, Tong et al. [4] reported that the wear resistance of the Al$_x$CoCrCuFeNi was similar to that of ferrous alloys at the same hardness level. Both these authors correlated the high

wear resistance to the higher hardness coming from the solid solution strengthening of single-phase HEAs. Hsu et al. [5] in their work discovered that the major wear mechanism of the $AlCoCrFe_xMo_{0.5}Ni$ HEA is abrasion. Also, by performing the oxidation test at the pin/disk interface flash temperature, 500 °C, they came to the conclusion that the oxidation rate of $Fe_{2.0}$ markedly exceeds that of $Fe_{1.5}$, indicating more oxides abrade the surface, resulting in lower wear resistance. Oxidative wear was also encountered by Du et al. [6] who studied the tribological behavior of the $Al_{0.25}CoCrFeNi$ HEA with a simple FCC phase and hardness of 260 HV from room temperature to 600 °C. They found out that below 300 °C, with increasing temperature, the wear rate increased due to high temperature softening. The wear rate remained stabilized above 300 °C due to the anti-wear effect of the oxidation film on the contact interface. The dominant wear mechanism of HEA changed from abrasive wear at room temperature to delamination wear at 200 °C, then delamination wear and oxidative wear at 300 °C and became oxidative above 300 °C. Excellent anti-oxidation property and resistance to thermal softening were reported for the $Co_{1.5}CrFeNi_{1.5}Ti$ and $Al_{0.2}Co_{1.5}CrFeNi_{1.5}Ti$ alloys, which are the main reasons for the outstanding wear resistance, which is at least two times better than that of conventional wear-resistant steels with similar hardness, such as SUJ2 and SKH51 [7].

Löbel et al. [8] investigated the wear behavior of $AlCoCrFeNiTi_{0.5}$ HEA produced by powder metallurgy under reciprocating wear conditions from room temperature to 900 °C. They found out that with increasing temperature up to 650 °C, initially, a slight decrease in wear resistance occurred, whereas a further increase in test temperature resulted in a distinct increase in wear resistance and a decrease in coefficient of friction. Their investigations prove the suitability of the $AlCoCrFeNiTi_{0.5}$ HEA for high-temperature applications, as the formation of protective oxides improves the wear performance.

A novel $AlCoCrFeNi_{2.1}$ eutectic CCA firstly studied by Lu et al. [9], is an alloy with promising properties in as-cast condition [10,11] due to the contribution of both ductile FCC ($L1_2$) and harder BCC (B2) phases resulting in a combination of good strength and ductility. The alloy has also shown to have excellent thermomechanical processing capability by severe cold rolling [12–15] or rotary friction welding [16]. Excellent work on surface wear and corrosion behavior of the $AlCoCrFeNi_{2.1}$ has been performed by Hasannaeimi et al. [17], where a transition from adhesive to oxidative wear was observed as the duration of reciprocating wear test increased.

The purpose of the present research is to study the tribological behavior of $AlCoCrFeNi_{2.1}$ eutectic CCA as a function of microstructure evolution by aging the material at 800 °C for 100, 500, and 1000 h, both for the sake of fundamental scientific understanding and the application worthiness of this particular alloy. Although the precipitation is the main process taking place during aging at 800 °C, the focus of the present research will be only on the effect these precipitates have on the wear behavior of the $AlCoCrFeNi_{2.1}$. Detailed analysis of precipitate kinetics, thermodynamics, and structure/morphology will be published elsewhere.

## 2. Materials and Methods

The eutectic $AlCoCrFeNi_{2.1}$ CCA was synthesized by vacuum induction melting and ingot casting using commercially available elements with a purity of 99.9%. Before melting, the furnace was evacuated and a subsequent argon gas atmosphere of 300 mbar was created. The molten material was kept for at least 20 min at 1500 °C under induction current, which provided sufficient agitation to ensure the homogeneous distribution of elements in the melt. The melt was cast inside the furnace (Ar atmosphere) in a cast-iron crucible and further cooled down in the air. In order to determine the melting point of the alloy and to find out what reactions might take place during the heating/cooling process, differential scanning calorimetry (DSC) analysis using STA 449 C Jupiter Thermo-microbalance (Netzsch-Gerätebau GmbH, Selb, Germany) was subsequently performed in the temperature range of 30–1425 °C under the dynamic atmosphere of Ar (15 mL $min^{-1}$), heating and cooling rate of 20 K $min^{-1}$, using an $Al_2O_3$ crucible with approximately 219.6 mg of sample material. As indicated by the DSC curves in Figure 1, besides melting and solidification peaks with onset temperatures 1343.9 °C and 1346.3 °C, respectively, a transformation reaction revealed in form of weak peaks can be seen on the

heating and cooling curves with onset temperatures around 800 °C and 822 °C, respectively. Therefore, in order to be able to characterize the phase transformation developing around 800 °C, dilatometry technique using Bähr DL 805A/D dilatometer (TA Instruments, Inc, New Castle, DE, USA) was used to heat a cylindrical sample with a diameter of 4 mm and a length of 10 mm to a temperature of 800 °C at which it was held for 20 h, then quenched by streaming $N_2$ gas to ensure a controlled cooling rate of 10 K $s^{-1}$. Afterward, the sample was longitudinally cut in half and a metallographic specimen was prepared by hot mounting in Bakelite, then mechanically ground with silicon carbide emery paper from a grade 180 down to 1200, and finally polished with 3 µm and 1 µm abrasive diamond suspensions. Scanning electron microscopy (SEM; JSM-6500F, Jeol, Tokyo, Japan) with back-scattered (BSE) detector and energy dispersive X-ray spectroscopy (EDS) were used for imaging and chemical composition determination. In addition, a lamella for transmission electron microscopy (TEM; JEM-2100 HR, Jeol, Tokyo, Japan, operated at 200 kV) was prepared for detailed microstructural characterization. A TEM lamella was first prepared by cutting a small sample of around 3 mm length, 1 mm height, and 1 mm thickness from a bulk specimen, then coarse and fine ground and polished to around 100 µm thickness using SiC papers from 800 to 4000 grit sizes, and additionally thinned by argon ion-slicing (IonSlicer, EM-09100IS, Jeol, Tokyo, Japan) with ion milling at 6 kV for around 6 h and around 15 min of fine ion milling at 2 kV to reach electron transparency. Scanning transmission electron microscopy (STEM) unit with a bright-field (BF) detector and EDS (JED-2300T, Jeol, Tokyo, Japan) were used for chemical composition determination and elemental mapping. Based on the characterization results, it was decided for the present work to isothermally age the material at 800 °C in a simple furnace (air atmosphere) at three different durations, 100, 500, and 1000 h and subsequently quench it in water.

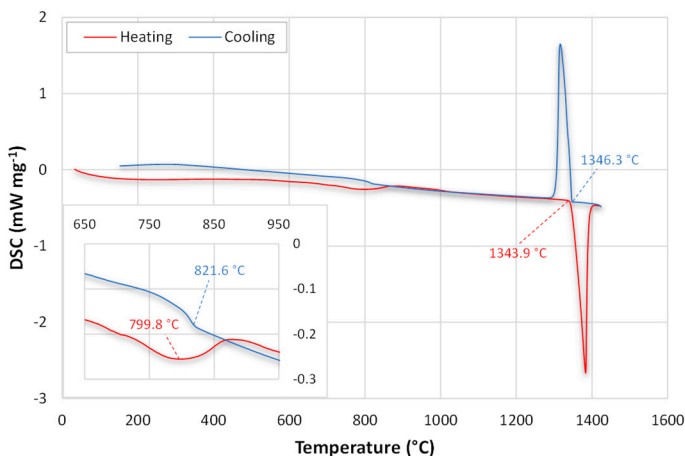

**Figure 1.** Heating and cooling curves from DSC analysis on the AlCoCrFeNi$_{2.1}$ CCA.

On as-cast and aged material, metallographic samples of cuboid shape with dimensions $20 \times 20 \times 10 \, mm^3$ were machined, ground and polished applying the same method as described above. SEM-BSE imaging and EDS analyses on at least 5 randomly chosen regions at both 1k and 3k magnifications were acquired. For more representative analysis, images acquired at 1k magnification were used for quantitative evaluation of phase fractions, whereas those acquired at 3k were used for quantitative evaluation of precipitates. Images were digitally analyzed using FIJI (ImageJ, ver. 1.52p, Bethesda, MD, USA) [18,19] with an appropriate size filter and color threshold, which enabled a separate analysis of the three distinct phases. As a result, the surface area of the darker phase, as well as surface area and distribution (*x* and *y* coordinates) of each precipitate could be obtained. Having these data and a known surface area of BSE images, the area fraction of all phases could be easily determined, where average values with standard deviations were derived out of five analyzed images. Furthermore, on the metallographic samples (cuboid $20 \times 20 \times 10 \, mm^3$), HV10 hardness measurements were performed at room temperature with at least three indentations and a holding time of 14 s using Instron Tukon 2100B instrument (Buehler-Illinois Tool Works (ITW), Lake Bluff, IL, USA). In addition, three repetitions

of reciprocating ball-on-plate wear tests using hardened DIN 100Cr6 bearing steel ball with a diameter of 20 mm as a counter-body were performed. The hardened bearing steel ball was used due to its high hardness (~700 HV) thus concentrating the major amount of wear on the investigated alloy's surface, as well as due to the similarity to many machine component applications with the prevailing metal-metal contact. Furthermore, the use of the steel counter-body enables the study of not only abrasive but also adhesive wear mechanisms.

Wear tests were performed using the in-house designed ball-on-plate reciprocating sliding device (Figure 2), as typically found in many tribological investigations, with a stationary ball (counter-body) being loaded against a moving flat-surface specimen (cuboid $20 \times 20 \times 10$ mm$^3$). Tests were performed at ambient temperature conditions ($21 \pm 2$ °C) with a sliding frequency of 15 Hz at a stroke of 4 mm, resulting in a maximal sliding velocity of 0.12 m s$^{-1}$ and a total duration of 833 s, which corresponds to a 100 m of sliding. The normal load of 20 N was applied, corresponding to 1 GPa of mean Hertzian contact pressure. All tests were performed in dry sliding conditions.

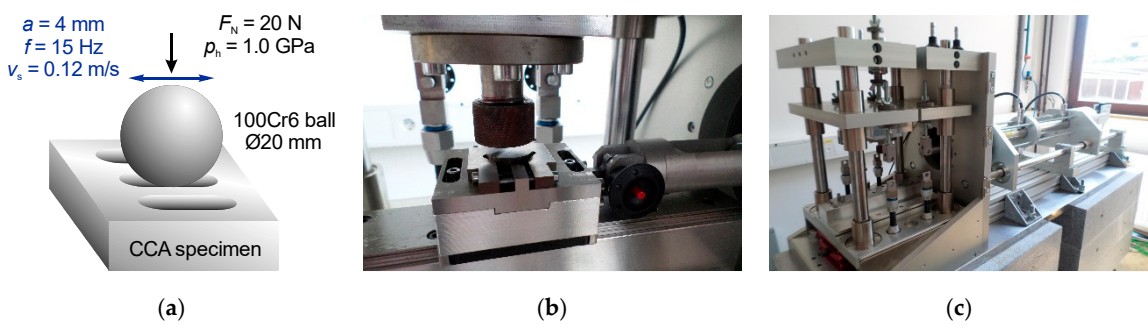

(a)　　　　　　　　　　　　　　(b)　　　　　　　　　　　　　　(c)

**Figure 2.** Ball-on-plate configuration used for wear tests: (**a**) Schematics with test parameters; (**b**) Configuration in the actual testing rig; (**c**) The device used for wear tests.

Worn specimens and counter-bodies were characterized using the 3D surface measurement system (Alicona InfiniteFocus, Alicona Imaging GmbH, Raaba, Austria), being able this way to accurately measure the worn-out material volume $V$ (mm$^3$) and calculate the specific wear rate coefficient ($k$) according to the equation

$$k = V \cdot F^{-1} \cdot s^{-1} \tag{1}$$

where $F$ (N) is the maximum load applied, and $s$ (m) is the total sliding distance.

Finally, SEM imaging and EDS analyses were performed on wear tracks in order to shed some light on wear mechanisms.

## 3. Results and Discussion

### 3.1. Microstructure and Hardness

The microstructure of the as-cast alloy with typical lamellar/dendritic morphology is shown in the backscattered SEM image in Figure 3a. Precipitation of a fine darker phase inside L1$_2$ (FCC) (light gray area) after 100, 500, and 1000 h of aging at 800 °C is shown in backscattered SEM images in Figure 3b–d, respectively. Besides precipitation and a slight coarsening/decomposition of the B2 lamellae, there are no major morphological changes of the lamellar/dendritic structure with aging. However, elemental concentrations within both phases slightly change with aging time, as shown in Figure 4.

Figure 4 shows the change in at% concentration of elements in all three phases, obtained by EDS analysis. A decrease of Al and Ni concentrations and an increase of the concentrations of Co, Cr, and Fe in the L1$_2$ (FCC) phase (Figure 4a) is more pronounced in the first 100 h of aging at 800 °C, as the precipitation kinetics is the fastest in this time span. The opposite can be seen in the B2 (BCC) phase (Figure 4b). This is in good agreement with changes in phase fraction and precipitation process (Figure 5), where precipitates size and area fraction increase at the expense of the area fraction of L1$_2$

phase, while the B2 area fraction also shows a slight increase. Furthermore, there is a noticeable decrease in at% concentration of Co, Cr, and Fe in the precipitates after 1000 h of aging at 800 °C (Figure 4c) and an increase of Al and Ni concentrations. Note also the similarity of chemical composition between the B2 (BCC) phase after aging (Figure 4b) and the precipitates phase after 1000 h of aging (Figure 4c). This is a clear indication of similarity of B2 and precipitates phase also from the viewpoint of the crystal lattice structure, which needs to be confirmed by additional analysis using XRD, selected area diffraction in TEM, or even high-resolution TEM for a detailed analysis of the interface between precipitates and $L1_2$ phase. This is beyond the scope of the present study and will be investigated separately.

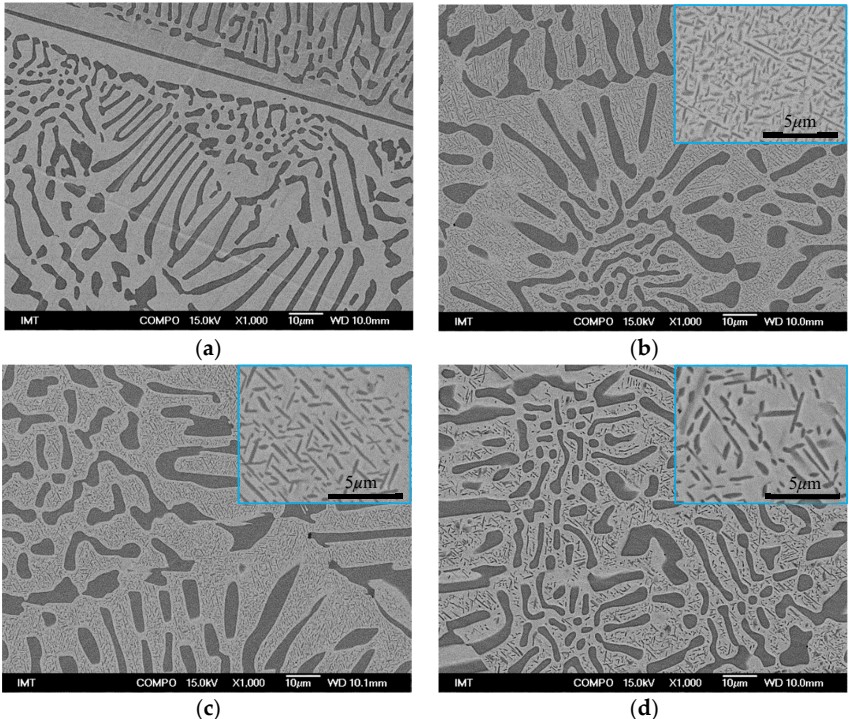

**Figure 3.** SEM-BSE images of the microstructure of: (**a**) As-cast AlCoCrFeNi$_{2.1}$ CCA with B2 (BCC) lamellae—dark grey and L1$_2$ (FCC) phase—light grey; Precipitation and coarsening of fine B2-like phase inside L1$_2$ after aging at 800 °C for: (**b**) 100 h; (**c**) 500 h; (**d**) 1000 h.

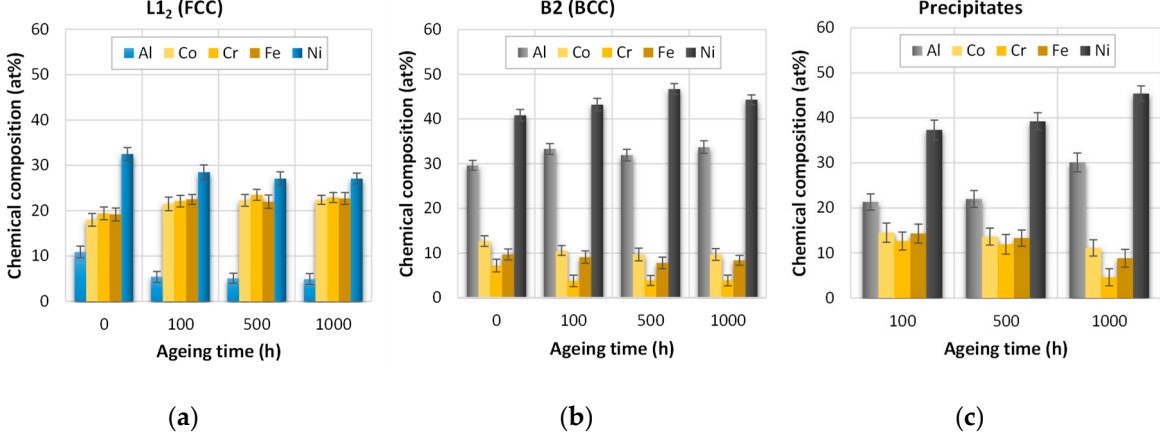

**Figure 4.** Changes of elemental concentration in at% with aging at 800 °C in: (**a**) L1$_2$ (FCC) phase; (**b**) B2 (BCC) phase; (**c**) Precipitates.

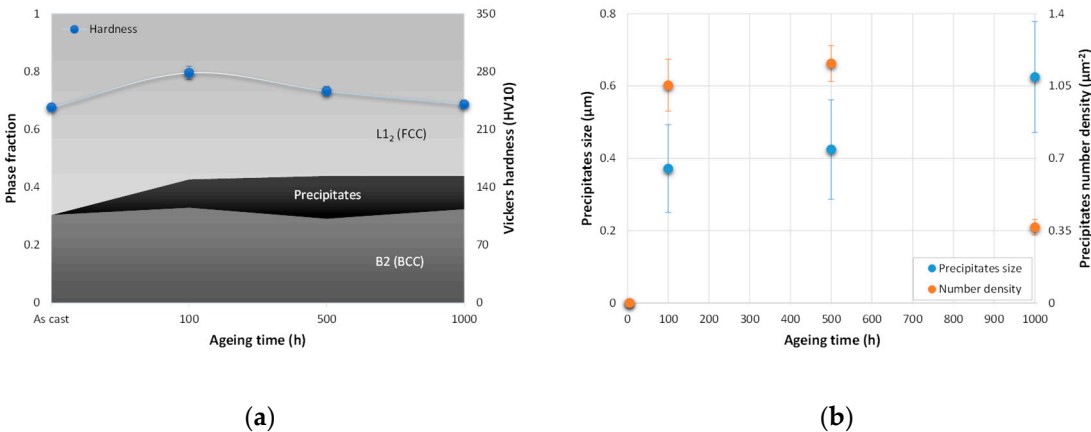

**Figure 5.** Microstructure evolution and hardness change as a function of aging at 800 °C: (**a**) Phase fractions and Vickers hardness; (**b**) Precipitates size and their number density.

Changes in phase fraction and Vickers hardness with aging are shown in Figure 5a, whereas precipitate size and number density are given in Figure 5b. The initial phase fractions of B2 and $L1_2$ phases is around 0.3 and 0.7, respectively. After the first 100 h of aging, the precipitates phase fraction (0.1) increases at the expense of the $L1_2$ phase that drops to 0.57, while B2 phase fraction also shows a slight increase (0.33). Precipitates average size expressed as equivalent circle diameter (ECD) at this stage is 0.37 ± 0.1 μm. From this point up to the 1000 h of aging, the $L1_2$ phase fraction remains virtually unchanged, whereas the phase fraction of precipitates slightly fluctuates between 0.15 and 0.12 in accordance with the B2 phase after 500 and 1000 h of aging, respectively. However, this does not mean that the $L1_2$ phase has reached its solid solution equilibrium state because the precipitation from the $L1_2$ phase continues even after 100 h of aging. The precipitate size continues to increase to 0.42 ± 0.1 μm and 0.62 ± 0.2 μm after 500 and 1000 h of aging, respectively. The precipitation rate is the highest in the first 100 h of aging, as represented by the number of precipitates per unit area or number density of precipitates, which at this stage is around 1.05 ± 0.13 $μm^{-2}$. The precipitation continues up to 500 h of aging but a with much lower rate, as the existing precipitates at this stage continue growing. From this point up to 1000 h of aging, the number of precipitates drops down to 0.37 ± 0.04 $μm^{-2}$. Since at the same time precipitates size increases while their number density decreases, it means that by aging the $AlCoCrFeNi_{2.1}$ at 800 °C from 500 h onward, the Ostwald ripening process develops, where larger particles coarsen at the expense of dissolving smaller ones.

The slight increase in hardness of $AlCoCrFeNi_{2.1}$ after the first 100 h of aging at 800 °C can be attributed to precipitation of the fine B2-like phase at the expense of the softer $L1_2$ phase, the fraction of which decreases at this stage (Figure 5a). From this point up to 1000 h of aging, the hardness decreases linearly back to the initial value (as-cast condition), which can be attributed to the precipitate coarsening within the softer $L1_2$ phase and continuous impoverishing of the solid solution from solute atoms such as Al and Ni, which diffuse towards the precipitates and the B2 phase.

STEM-EDS elemental analysis and mapping shown in Figure 6b and summarized in Table 1, indicate the different composition of the B2 phase in spectra 1 and 2, and the $L1_2$ phase in spectrum 3 [20]. Spectrum 2 in Table 1 and the elemental map for Cr, at the lower right corner shows, increased Cr signal, which is coming from the Cr-rich nano-precipitates, as reported by Gao et al. [11]. Similar to the B2 phase (spectra 1 and 2), Al and Ni concentration in precipitates (spectra 4, 5, and 6) is higher as compared to the $L1_2$ phase (spectrum 3). A detailed analysis of these precipitates from the viewpoint of their crystal lattice structure, kinetics, thermodynamics of precipitation, etc., is beyond the scope of the present work, therefore it will be studied separately elsewhere.

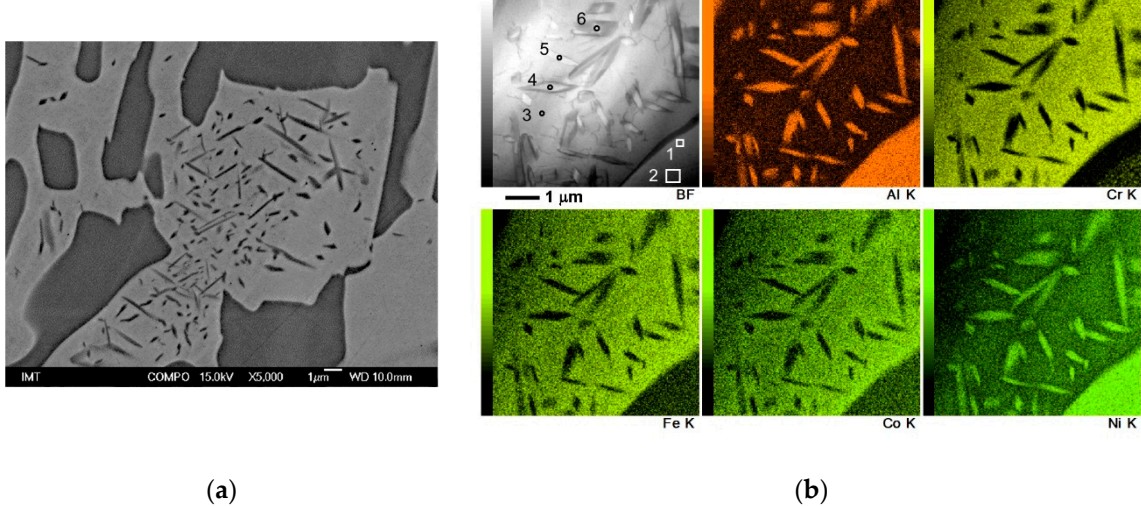

<div style="text-align:center">(<b>a</b>)　　　　　　　　　　　　　　　　　　　　(<b>b</b>)</div>

**Figure 6.** Electron microscopy characterization of the dilatometry sample (20 h at 800 °C): (**a**) SEM-BSE image showing partial precipitation; (**b**) STEM bright-field image of precipitates with EDS point analyses and mapping.

**Table 1.** Elements concentration in at% from EDS point analyses in Figure 6.

| Spectrum | Al | Co | Cr | Fe | Ni | Total |
|---|---|---|---|---|---|---|
| 1 | 13.4 | 13.8 | 4.9 | 9.4 | 58.4 | 100 |
| 2 | 13.2 | 12.7 | 8.7 | 10.8 | 54.6 | 100 |
| 3 | 3.5 | 18.3 | 22.6 | 20.4 | 35.3 | 100 |
| 4 | 19.8 | 10.1 | 3.7 | 8.7 | 57.8 | 100 |
| 5 | 18.6 | 11.1 | 5.2 | 9.7 | 55.3 | 100 |
| 6 | 18.6 | 11.1 | 4.2 | 9.5 | 56.6 | 100 |

### 3.2. Wear Tests

A representative example of 1 out of 3 wear tracks of tested samples is shown in Figure 7 in form of 3D pseudo-color depth images, 2D optical images, and cross-sectional depth profiles of all three wear tracks taken at the same locations shown with the red line across the representative wear tracks in Figure 7a–d. Visual differences in shape or length/width/depth of wear tracks depending on materials condition (aging time) are minor, however, detailed volume measurements could reveal their difference.

The coefficient of friction (COF) shown in Figure 8 is characterized by both short- and long-range fluctuations. The short-range (high frequency) fluctuations can be attributed to the difference in coefficient of friction between the harder B2 phase and the softer $L1_2$ phase. This is supported by the findings of Hasannaeimi et al. [17] who evaluated a small-scale phase-specific scratch behavior and found the variation in COF across the scratch line, where the softer FCC phase shows lower COF as compared to the harder BCC one. Long-range fluctuations, however, could be coming from the different morphology of the new contacting surface, which is continuously uncovered by wearing out the top-most layer of the material in dry sliding contact.

The trend at which the average coefficient of friction of the as-cast $AlCoCrFeNi_{2.1}$ CCA changes with aging time as shown in Figure 9, is in a good correlation with the changing trend of $L1_2$ + precipitates phase fraction. This suggests that the plastic deformation of the softer $L1_2$ matrix with precipitates results in higher COF than the harder B2 phase. In general, by aging the $AlCoCrFeNi_{2.1}$ CCA at 800 °C, COF of decreases with time.

A trend opposite to the one in Figure 9 is shown in Figure 10 for the specific wear rate of the specimen and counter-body, as well as the B2 phase fraction. In as-cast material, the wear rate of counter-body is lower compared to the wear rate of the material. After 100 h of aging at 800 °C, the wear rates of both specimen and counter-body increase and become almost equal. At this point, the B2 phase fraction and hardness also increase slightly, whereas COF decreases. After 500 h of aging, the specific wear rate of the specimen decreases almost three times more than the wear rate of counter-body (Figure 10), whereas the COF remains virtually unchanged (Figure 9). At this point, there is a slight decrease in B2 fraction and a slight increase in precipitate size, fraction, and number density (Figure 5). With further aging up to 1000 h, the wear rate of specimen rises almost to the value of the as-cast state but still remains lower than the wear rate of the counter-body. From this analysis, it can be concluded that adequate aging of AlCoCrFeNi$_{2.1}$ CCA at 800 °C not only decreases the coefficient of friction but also improves the material's wear performance.

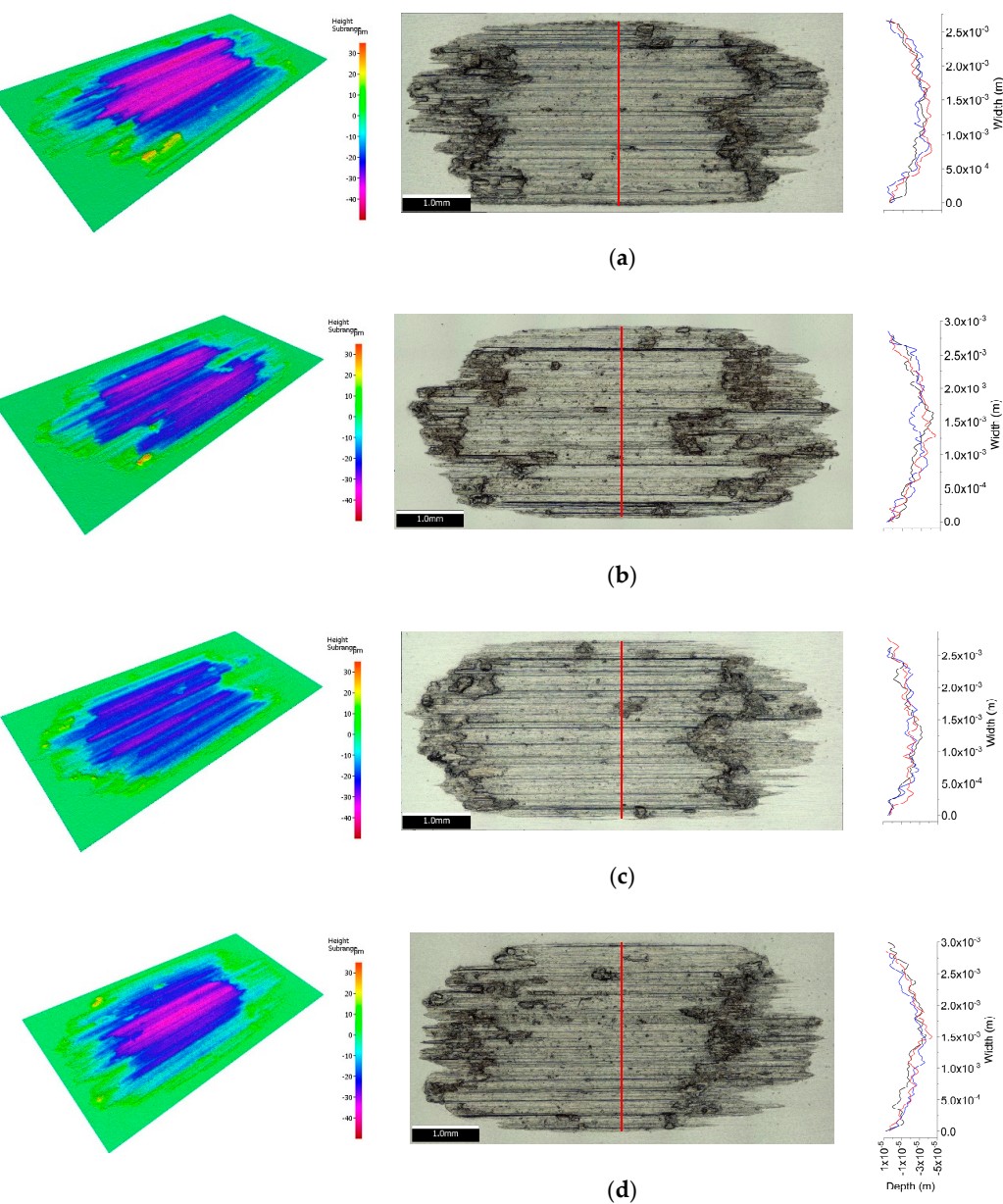

**Figure 7.** Pseudo-color 3D depth images, optical images, and depth profiles of three wear tracks in (**a**) As-cast material; Material aged at 800 °C for (**b**) 100 h; (**c**) 500 h; (**d**) 1000 h.

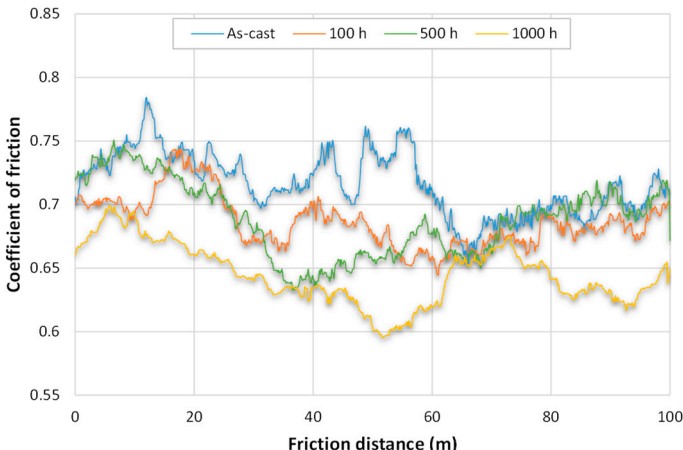

**Figure 8.** Coefficient of friction throughout the wear test on as-cast AlCoCrFeNi$_{2.1}$ and after aging at 800 °C at three different durations.

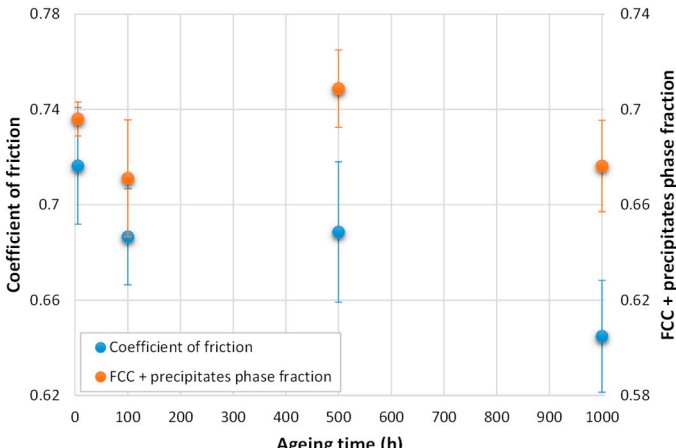

**Figure 9.** Coefficient of friction and FCC (L1$_2$) + precipitates phase fraction of as-cast AlCoCrFeNi$_{2.1}$ and after aging at 800 °C at three different durations.

SEM-BSE images of the wear track of as-cast AlCoCrFeNi$_{2.1}$ CCA shown in Figure 11a reveals the presence of patches of re-deposited material caused by adhesive wear, erosion grooves from abrasive wear, and severe disintegration and some delamination of the lamellar morphology inside the wear track. An SEM-BSE image at higher magnification in Figure 11b indicates extensive plastic deformation through the planar slip mechanism of the L1$_2$ phase. When the planar slip lines in the L1$_2$ phase reach the semi-coherent interface between the latter and B2 phases, a step-like shape is formed, as shown in Figure 11b.

Inside the wear track, both phases deform uniformly and no inter-phase detaching or void formation is observed. Gao et al. [11] reported that the L1$_2$ phase can accommodate several arrays of parallel mobile dislocations and deform by planar slip. They also reported that the B2 phase in the eutectic fails in a brittle manner, while the L1$_2$ phase shows ductility and necking, leading to dual-mode fracture in this alloy. Hasannaeimi et al. [17] reported that during wear test of the AlCoCrFeNi$_{2.1}$, L1$_2$ and B2 phases deformed simultaneously, while the B2 phase accommodates medium density of dislocations. This was attributed to the 3D back-stress acting on the L1$_2$ phase which can maintain synchronous deformation in heterogeneous systems. This back stress is further enhanced by semi-coherent boundaries between B2 and L1$_2$ phases, and the lower fraction of B2 lamellae. The accumulative effect of these conditions resulted in the activation of dislocation in the brittle B2 phase facilitated by a high density of dislocation pile-up at the phase boundaries, and this modified the brittle behavior of B2 phase to accommodate deformation.

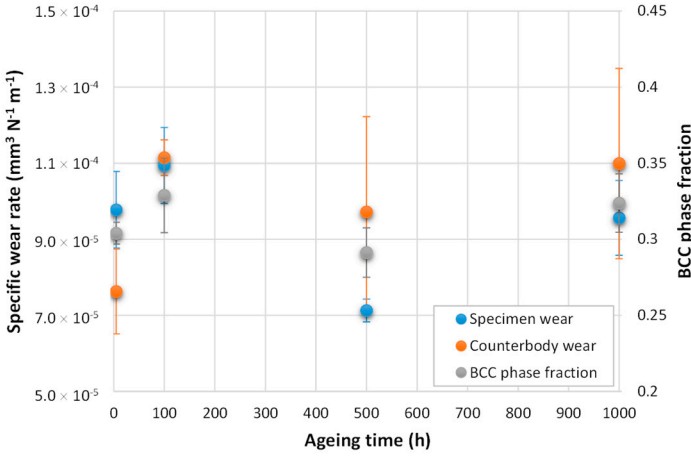

**Figure 10.** The specific wear rate of specimens and counter-body, and BCC (B2) phase fraction of as-cast AlCoCrFeNi$_{2.1}$ and after aging at 800 °C at three different durations.

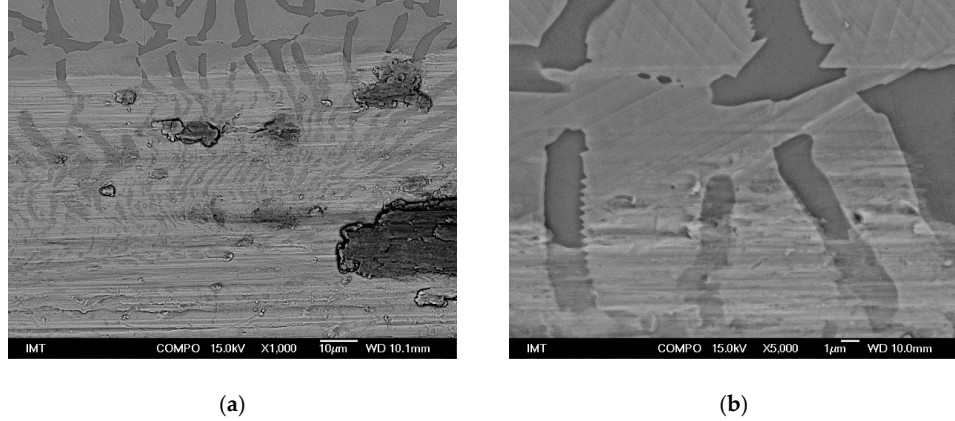

| (a) | (b) |

**Figure 11.** SEM-BSE images of wear tracks of AlCoCrFeNi$_{2.1}$ CCA in as-the cast state: (**a**) Wear track edge with patches of re-deposited material; (**b**) Slip lines in the vicinity of wear track.

SEM-EDS analyses in Figure 12a summarized in Table 2, indicate that inside the wear track, the main wear mechanism is abrasion with little patches of oxidized redeposited material (self-adhesion) and some surface delamination. Figure 12b shows EDS analyses of the piled-up oxides of counter-body and self-adhered material, as summarized in Table 3, where Si and Mn, which are present in the counter-body material (DIN 100Cr6) are also detected.

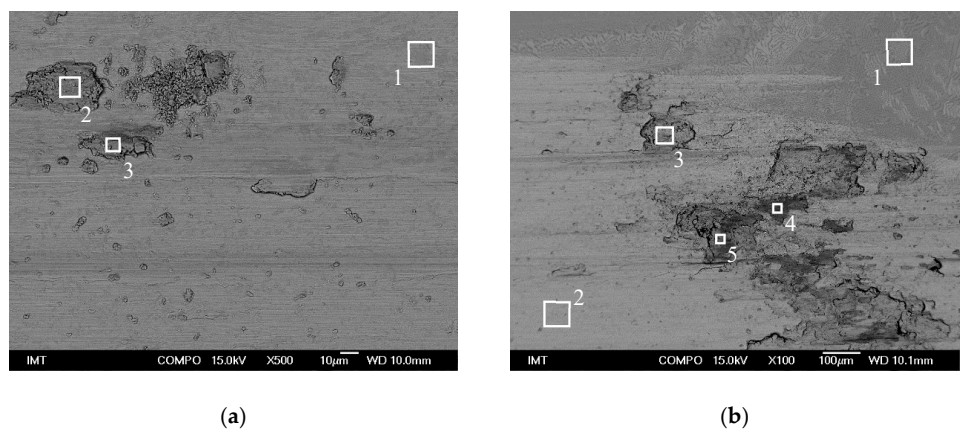

| (a) | (b) |

**Figure 12.** SEM-BSE images with EDS analyses on the wear track of AlCoCrFeNi$_{2.1}$ in the as-cast state: (**a**) Inside the wear track; (**b**) At the edge of the wear track.

Similar oxide pile-up phenomenon but in reduced amount is present in all durations of the aged material, as shown in Figure 13a for 100 h of aging. The difference between the aged material and the one in as-cast condition is the fraction of $L1_2$ and B2 phases, resulting in different hardness level, as well as the presence, size, and number density of precipitates that strengthened the $L1_2$ phase, resulting in no plastic deformation by planar slip, as seen in the as-cast material. This is shown in Figure 13b for 100 h of aging and is similar to the material aged for 500 and 1000 h. Differences observed in tribological behavior obtained by aging can be ascribed to these changes, causing alterations in wear mechanism. In the case of as-cast alloy without precipitates, due to lower B2 phase fraction as compared to the $L1_2$, abrasive wear is predominantly combined with adhesive wear, where delamination is largely prevented by dislocation pile-up and planar slip at the phase boundaries [11], thus modifying the brittle behavior of B2 phase [17]. Intensified adhesive wear component results in higher friction but low counter-body wear. After 100 h of aging, the adhesive wear component is reduced due to lower $L1_2$ phase fraction and increased hardness, also indicated by a small drop in friction. However, increased B2 phase fraction, absence of planar slip, and precipitation strengthening result in increased brittleness and intensified delamination. This amplifies the formation of hard wear particles, which remain in the reciprocating sliding contact and lead to increased wear of the material and counter-body (Figure 10). Prolonged aging time (500 h) results in a slightly reduced B2 phase and increased $L1_2$ phase fractions but a high level of strengthening with precipitates (Figure 5b), thus providing the best combination of high hardness and reduced brittleness with minimal wear. In this case, the main wear mechanism is abrasive wear with minimized components of adhesion and delamination. Furthermore, over-aging for 1000 h leads to a decrease in the number density of precipitates and an increase in their size, which results in decreased hardness, as shown in Figure 5. Lower hardness means increased abrasive wear, with a smaller number of large precipitates representing reduced resistance to sliding and a thus lower coefficient of friction (Figure 9).

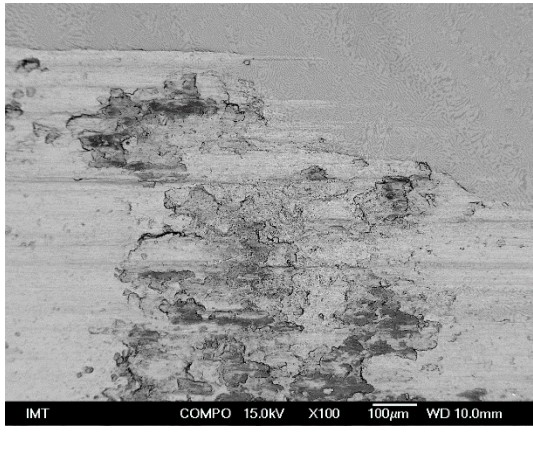
(**a**)

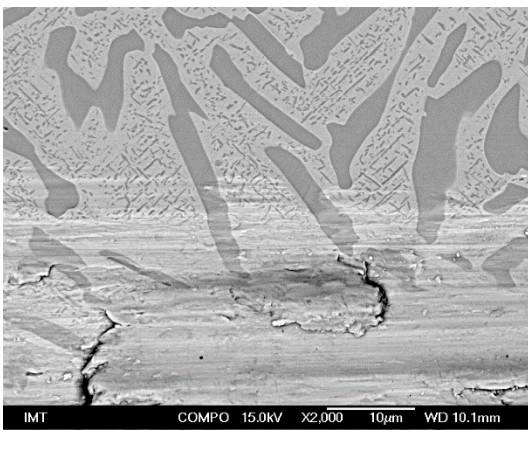
(**b**)

**Figure 13.** SEM-BSE images of the wear track edge after 100 h of aging at 800 °C: (**a**) Oxide pile-up at the edge of the wear track; (**b**) Edge of the wear track showing no deformation by planar slip inside the $L1_2$ phase.

**Table 2.** Elements concentration in at% from EDS point analyses in Figure 12a.

| Spectrum | O | Al | Cr | Fe | Co | Ni | Total |
|---|---|---|---|---|---|---|---|
| 1 | 0.0 | 24.1 | 12.3 | 13.0 | 12.7 | 37.8 | 100 |
| 2 | 29.0 | 9.6 | 11.6 | 20.2 | 9.4 | 20.1 | 100 |
| 3 | 23.9 | 8.7 | 13.4 | 24.9 | 9.3 | 19.9 | 100 |

**Table 3.** Elements concentration in at% from EDS point analyses in Figure 12b.

| Spectrum | O | Al | Si | Cr | Mn | Fe | Co | Ni | Total |
|---|---|---|---|---|---|---|---|---|---|
| 1 | 1.9 | 16.2 | 0.0 | 15.1 | 0.0 | 15.9 | 16.6 | 34.3 | 100 |
| 2 | 0.0 | 15.6 | 0.0 | 15.9 | 0.0 | 16.6 | 17.1 | 34.8 | 100 |
| 3 | 15.0 | 3.3 | 0.6 | 16.3 | 0.9 | 48.9 | 3.5 | 11.5 | 100 |
| 4 | 47.6 | 3.0 | 0.4 | 9.9 | 0.5 | 26.4 | 3.8 | 8.4 | 100 |
| 5 | 44.1 | 4.2 | 0.5 | 10.7 | 0.4 | 25.7 | 4.4 | 10.0 | 100 |

## 4. Conclusions

In the present study, wear behavior as a function of aging time was evaluated for the $AlCoCrFeNi_{2.1}$ eutectic complex, concentrated alloy consisting of B2 (BCC) lamellae, and $L1_2$ (FCC) phase in as-cast state, as well as fine evenly distributed precipitates inside the $L1_2$ phase after aging at 800 °C for 100, 500, and 1000 h. Between 0 and 500 h of aging, both precipitates number and size increase, while from 500 h to 1000 h of aging, precipitates coarsen by Ostwald ripening mechanism. Abrasive wear prevailed in reciprocating wear analysis, with adhesive wear and delamination component changing depending on the aging conditions. The $L1_2$ phase without precipitates and lower hardness in the as-cast material preferentially deformed during the wear test, resulting in intensified adhesive wear component but minimum delamination, hindered by planar slip between B2 and $L1_2$ phases. This was not the case after aging the material, i.e., with the presence of precipitates. In general, aging the $AlCoCrFeNi_{2.1}$ alloy at 800 °C decreases its coefficient of friction due to reduced adhesive wear component and enhances its wear performance through precipitation strengthening. However, under-aging (100 h) results in increased material brittleness and thus increased delamination, while over-aging (1000 h) results in precipitates coarsening, and decreased hardness and abrasive wear resistance. The best performance, combining high hardness and reduced brittleness with minimal wear, is achieved with an intermediate aging of 500 h.

**Author Contributions:** Conceptualization, F.K. and B.P.; Methodology, F.K.; Formal analysis, F.K.; Investigation, F.K. and D.F.; Writing—original draft preparation, F.K.; Writing—review and editing, B.P. and D.F.; Visualization, F.K.; Project administration, F.K.; Funding acquisition, F.K. All authors have read and agreed to the published version of the manuscript.

**Funding:** This research was funded by Javna Agencija za Raziskovalno Dejavnost Republike Slovenije, grant number Z2-9220.

**Acknowledgments:** Acknowledgements go to the Institute of Metals and Technology (IMT) in Ljubljana where research was performed. Also, authors are thankful to all who were involved in the experimental part of this research.

**Conflicts of Interest:** The authors declare no conflict of interest. The funders had no role in the design of the study; in the collection, analyses, or interpretation of data; in the writing of the manuscript, or in the decision to publish the results.

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
