# Peer review of "Tribological Behavior of As-Cast and Aged AlCoCrFeNi2.1 CCA"

_metals, doi:10.3390/met10020208_

Round 1

Reviewer 1 Report

General Comments

The manuscript discusses the wear behavior as a function of ageing time which was evaluated for the AlCoCrFeNi2.1 eutectic complex. The paper is obviously of interest for researchers working in this field. However, the manuscript should be amended before its acceptance. I would like to address the following issues:

The ‘Materials and Methods’ section requires expanding. A lot of important information on hardness measurements is missing (what equipment was used, how many indents were made, what was the load hold time). In addition, there are no details of the device on which the tribological tests were performed, and no justification for the test conditions or for the selection of the 100Cr6 bearing steel as the counter-specimen. There is no information provided on how the specific wear rate was determined (including the formula). Figure 5 shows a representative example of wear tracks of the specimen and the ball. It is too general. Please present similar figures for the other test variants. In the description of Fig. 6 the term “friction distance” would be more suitable than “test duration”. The results of the tribological tests require a more in-depth discussion. In the graphs shown, e.g. after 100 h of ageing, the material had the highest hardness and yet the wear was the highest. Similarly, after 1000 h of ageing, the lowest friction coefficient was found, while tribological tests showed a significant increase in the specific wear rate, etc. The obtained test results are very poorly correlated with the results of research carried out by other authors. The summary is too short and too general.

Author Response

Dear Reviewer 1,

We are truly grateful for taking the time to look over our manuscript and provide us with valuable constructive feedback. The clarity and quality of the revised text has certainly improved as a result.

Please find attached the detailed responses to the points that have been raised within the work. We have also provided a revised version of our manuscript for your consideration.

On behalf of all co-authors, sincerely,

Fevzi Kafexhiu

Reviewer 2 Report

The paper studies the reciprocating wear resistance of AlCoCrFeNi2.1 CCA regarding the microstructure state. Ageing treatments were conducted to provoke segregation effects. A clear influence is proven. Some additional information is required to ensure reproducibility of the study. Excluding the phase assignment, the assumptions and conclusions sound meaningful, whereas the presentation of the results need to be improved. Additional analyses can enhance the worth of the study. A major revision should be conducted to improve the manuscript.

General

Doesn’t CCA means “Compositionally Complex Alloys”? Further information about the measurement equipment are needed in a uniform way (model, company, town, country) What was the heating and cooling rate of DSC measurement? What atmosphere was used during DSC measurement? What is the basis of assessment of phase assignment? A clear reference is necessary. Additional XRD measurement is better if possible Without knowing the specific lattice parameters, how can a B2 and L12 type be assigned. Doesn’t the ageing certainly change the local chemical composition and can affect the lattice/structure? Which quantity of sample was produced? How many times remelted? The cooling condition need to be stated. FCC phase is supersaturated and grain size of “bcc” phase depend on this Cooling conditions seems to have a strong effect on local chemical composition of the different microstructure domains. In this context it is necessary to additionally determine the chemical composition of the as cast state. This can contribute to a better understanding of the segregation effects regarding the light grey phase. While it’s quite difficult to clearly determine small precipitates, the composition of the main phases can help. Additionally, chemical composition of fcc phase should be analysed with regard to the distance of bcc parts (perhaps line scan) Interpolation lines in the diagrams should be omitted – in general the diagrams style should be improved Figure 4 needs to be corrected – region 4 is marked twice Figure 5 need a scale x- and z-axis Figure 10 and Figure 11 should be taken with the same magnification to enhance the comparability

By line

Line 12 – space after 500
line 18 – cca have to be introduced in the abstract
line 30 – CCA is introduced twice in the main part
line 44-57 – The literature review on high temperature wear properties focusing a limited temperature range and need to be extended.  https://doi.org/10.3390/e21060582 explore the wear properties up to 900 °C including coefficient of friction. Up to 500 °C a slight increase in wear was found for reciprocating ball on plate test. By further increase in temperature level the wear decreases. This should be added.
Line 78 – which quantity were casted and what are the cooling condition?
Line 78 – it seems that 20h aging is the sample of Figure 4. 20h aging need to be stated in the description of Fig. 4. Why the precipitates partially reach a length of 10µm while longer aging reveal shorter precipitates. Does the casting condition differ?
line 81 – was only one heat treatment conducted after casting?
Line 88 – further information about the hardness measurement are required. phase specific hardness measurement using nanoindentation can deliver more worthful information
Line 127 – Fig 3 is a little bit confusing. Interpolation (lines) fig. 3 b should be omitted. What means the double designation precipitates size – precipitates fraction? Is the standard deviation the fraction like d10 to d90 value? The particle size curve implements an increase by duration of ageing treatment. Detail view of Fig. 2 c and d show the length decreases. Between Fig. 2 c and d a strong increase in contrast can be seen. Perhaps the picture analyse is faulty. It should be stated at which magnification the analysation was performed. Comparing tests at different magnifications are recommended.
Line 136 – 20 h treatment should be stated
Line 137 – at% values are better
Line 182 – the fluctuation of counter body wear should be disussed.
Line 182 – ageing time -100°C? – a different diagram type seems to be better
Line 204 – SEM image BSD?
Line 208,209 – at% seems to be better
Line 227 – space after 500

Author Response

Dear Reviewer 2,

We are truly grateful for taking the time to look over our manuscript and provide us with valuable constructive feedback. The clarity and quality of the revised text has certainly improved as a result.

Please find attached the detailed responses to the points that have been raised within the work. We have also provided a revised version of our manuscript for your consideration.

On behalf of all co-authors, sincerely,

Fevzi Kafexhiu

Round 2

Reviewer 2 Report

Thanks for taking into account remarks and suggestions. The manuscript can be published in the current form.